# Integration of Epidemiological and Genomic Data to Investigate H5N1 HPAI Outbreaks in Northern Italy in 2021–2022

**DOI:** 10.3390/pathogens12010100

**Published:** 2023-01-06

**Authors:** Diletta Fornasiero, Alice Fusaro, Bianca Zecchin, Matteo Mazzucato, Francesca Scolamacchia, Grazia Manca, Calogero Terregino, Tiziano Dorotea, Paolo Mulatti

**Affiliations:** Istituto Zooprofilattico Sperimentale delle Venezie, 35020 Legnaro, Italy

**Keywords:** HPAI, H5N1, Italy, genetic network, epidemiological investigation, contact tracing, ERGM

## Abstract

Between October 2021 and April 2022, 317 outbreaks caused by highly pathogenic avian influenza (HPAI) H5N1 viruses were notified in poultry farms in the northeastern Italian regions. The complete genomes of 214 strains were used to estimate the genetic network based on the similarity of the viruses. An exponential random graph model (ERGM) was used to assess the effect of ‘at-risk contacts’, ‘same owners’, ‘in-bound/out-bound risk windows overlap’, ‘genetic differences’, ‘geographic distances’, ‘same species’, and ‘poultry company’ on the probability of observing a link within the genetic network, which can be interpreted as the potential propagation of the epidemic via lateral spread or a common source of infection. The variables ‘same poultry company’ (Est. = 0.548, C.I. = [0.179; 0.918]) and ‘risk windows overlap’ (Est. = 0.339, C.I. = [0.309; 0.368]) were associated with a higher probability of link formation, while the ‘genetic differences’ (Est. = −0.563, C.I. = [−0.640; −0.486]) and ‘geographic distances’ (Est. = −0.058, C.I. = [−0.078; −0.038]) indicated a reduced probability. The integration of epidemiological data with genomic analyses allows us to monitor the epidemic evolution and helps to explain the dynamics of lateral spreads casting light on the potential diffusion routes. The 2021–2022 epidemic stresses the need to further strengthen the biosecurity measures, and to encourage the reorganization of the poultry production sector to minimize the impact of future epidemics.

## 1. Introduction

The end of the 2021 summer season was characterized by several highly pathogenic avian influenza (HPAI) outbreaks of the H5 and H5N1 subtypes in wild and domestic birds in western Russia, as well as close to the western and eastern borders with Kazakhstan and Mongolia countries. These areas are well-known autumn staging sites for wild water birds that later become ready to migrate to Europe for overwintering. The migration of infected wild birds towards the northern and eastern European countries has been considered one of the major risks for the introduction of avian influenza (AI) and its further spread to the southern and western areas of Europe, recurring almost every autumn/winter season [1]. In 2021–2022, Europe experienced the largest highly pathogenicity avian influenza (HPAI) epidemic ever observed, affecting a total of 37 European countries, with approximately 2500 outbreaks notified in poultry, 47.7 million birds culled, 187 outbreaks in captive birds, and around 3600 outbreaks in wild birds [1].

In the past three decades, Italy has been recurrently affected by the circulation of AI viruses, with epidemics of both HPAI and Low Pathogenicity Avian Influenza (LPAI). This is likely due to the high concentration of poultry farms in confined geographic areas (Densely Populated Poultry Areas—DPPAs), in close proximity to vast wetlands located along the Black Sea–Mediterranean and the East Atlantic migratory flyways, which may serve as wintering and nesting areas to many migratory wild waterfowl [2,3]. Despite the large epidemics of HPAI and LPAI that occurred in the late 1990s and early 2000s, only sporadic cases were observed until 2016, when an H5N8 HPAI virus was responsible for two consecutive epidemic waves, causing a total of 83 outbreaks in poultry farms and 14 in wild birds, mainly in the northern regions of Italy [4]. 

On 18 October 2021, an H5N1 HPAI virus was detected in a fattening turkey farm in Verona Province, in the very core of the DPPAs. In the following weeks, the number of confirmed outbreaks rapidly increased and the epidemic spread to the neighbouring provinces and regions. By April 2022, a total of 317 outbreaks in poultry farms and 23 in non-poultry birds (22 in wild birds and 1 in captive birds) were reported in nine different Italian regions [1]. The viral circulation mainly occurred within the DPPAs located between the Veneto and Lombardia Regions, and in particular, in the provinces of Verona, Padova, Vicenza, Brescia, and Mantova. Similarly to the 2017–2018 HPAI H5N8 epidemic [4], the AI viruses appeared to have been directly introduced as highly pathogenic strains. The epidemic predominantly affected fattening turkey farms, as commonly observed in past events [4,5,6]; however, unlike the epidemics experienced since 1999, the 2021–2022 epidemic was characterized by a marked involvement of the broiler sector, although often with scarce or no symptoms reported. Of the 23 confirmed cases reported in non-poultry birds, only a limited number of involved birds belonged to the Anseriformes order (considered as the target species for the passive surveillance activities to detect AI in wild populations) [1], while several cases were reported in species usually considered marginally affected by HPAI viruses, such as magpies and owls [7,8]. The phylogenetic analyses suggested that at least 12 different introductions into the domestic sector and wild populations occurred during the epidemic, indicating that lateral transmission events most likely happened, determining a very high number of secondary cases in the poultry sector [9].

Herein, we present an approach to integrate genomic and epidemiological data collected during the 2021–2022 HPAI H5N1 epidemic in Italy, applying techniques of network analysis to genetic networks. Such an approach allowed us to assess the effects of potential drivers of virus circulation on the emergence of secondary outbreaks via lateral spreads in domestic poultry premises. The results may provide insights on shortcomings in the management of an epidemic, and could be used to inform future prevention measures to limit the massive spread between farms in DPPAs.

## 2. Materials and Methods

### 2.1. Genomic Analysis and Network Construction

The complete genome of 342 Italian viruses was generated as previously described [4]. Virus sequences were deposited in the GISAID EpiFlu Database (https://platform.gisaid.org/, accessed on 31 August 2022) with the accession numbers reported in Appendix A. Sequences were aligned in MAFFT v7 [10] and compared to the most related sequences available in GISAID (Appendix A). Maximum likelihood phylogenetic trees of each gene segment were constructed by using IQTREE v1.6.6 (https://github.com/iqtree/iqtree1, accessed on 22 November 2022) and an ultrafast bootstrap resampling analysis (1000 replications). Phylogenetic trees were visualized in FigTree v1.4.4 (http://tree.bio.ed.ac.uk/software/figtree/, accessed on 22 November 2022). The datasets used to generate the phylogenetic trees included all the Italian viruses sequenced from October 2021 to April 2022.

The phylogenetic network was generated using the Median Joining (MJ) method implemented in NETWORK 10.2.0.0 [11] for the eight concatenated gene segments of non-reassortant H5N1 viruses, allowing us to visualize how viral genomes are connected on the basis of their genetic similarity. For this purpose, the phylogenetic network analysis was run on a dataset composed of 229 Italian non-reassortant H5N1 viruses (including viruses from domestic and wild birds, marked with a star in Appendix A); 9 H5N1 viruses from Poland; and 8 H5N1 viruses from the Czech Republic belonging to the same genetic cluster. The MJ network uses a maximum parsimony approach to reconstruct the relationships between highly similar sequences; it consists of nodes and links (also referred to as ties or edges) that connect the nodes. Two connected nodes within the network represent a dyad. The nodes can be either isolated viral strains or median vectors, which represent hypothesized sequences used to connect the existing viruses in the most parsimonious way.

From the hypotheses generated by the interpretation of the phylogenetic tree and the genetic network described above, a second MJ network was constructed using only a subset of the available viral strains isolated from the Italian poultry compartment (referred to as cluster A), containing only a single isolate per outbreak. This network was then used for the analysis to assess the impact of epidemiological drivers on the potential viral spread between poultry farms.

### 2.2. Network Covariates

The information collected through on-field epidemiological enquiries, contact tracing activities, and phylogenetic analyses were exploited to derive a series of factors to be further analyzed as covariates in the genetic network. All data cleaning, variable calculation, and statistical analysis operations were performed using the R [12] and RStudio [13] software. The raw data were processed to obtain three types of terms, with different information accordingly to the specific network feature to which they refer: (i) network density, (ii) nodal-level covariates, and (iii) edge-level covariates. Network density refers to the total number of edges observed in the network, and it is used to evaluate whether the observed network exhibits an intrinsic non-random structure that could be related to a series of other potential drivers. The nodal-level covariates encompass information associated with the nodes within the network, which are related to the outbreaks. Specifically, the nodal homophily (i.e., the tendency of two nodes to share the same attribute) was taken into account for the species reared and the poultry company, to include information on the similarity of the two end nodes (i.e., outbreaks) of a potential tie (i.e., highest genetic similarity) in rearing the same species and belonging to the same company. Therefore, two dichotomic matrices were built considering the pairwise comparisons of all the outbreaks, in order to indicate whether the compared outbreaks had the same condition (i.e., rearing the same species or belonging to the same poultry integrator). The edge-level covariates included a list of five pairwise matrices containing information characterizing each pair of outbreaks (i.e., dyadic relationship): (i) overlap between risk time windows; (ii) number of at-risk contacts; (iii) geographical distances; (iv) belonging to the same owner; and (v) genetic differences between the viruses identified at each infected farm. 

The ‘risk time windows overlap’ represents the period during which each farm was exposed to the potential infection spreading from other infected premises. It was determined considering the overlap of two at-risk time windows: an ‘out-bound risk window’ (ORW), representing the period in which the avian influenza virus (AIV) could have been spread from an infected farm, and an ‘in-bound risk window’ (IRW), which indicated the period in which an AIV could have been introduced into an infected farm. The ORW was estimated to range from four days prior to the onset of symptoms (to account for the period in which the birds can be infectious without showing symptoms [14]) until the extinction of the outbreak. The IRW was assumed to begin 15 days before the symptom onset, accounting for the viral incubation period [15], and ending on the day on which symptoms were observed. The ‘risk time windows overlap’ was calculated as the number of days in which the ORW and the IRW of each pair of outbreaks overlapped. This measure could range from a maximum of 16 days (corresponding to the full overlap between the two risk windows) to the minimum value of 0 when the two periods did not overlap.

As for the ‘at-risk contacts’, all the available data on the movement of vehicles and personnel (i.e., technicians and veterinarians) were entered into a structured dataset and analyzed through the R package EpiContactTrace [16], considering the previously described IRW and ORW. This allowed us to identify the total number of contacts occurred between farms that could have been considered productive in terms of infection spread. Geographic distances were included to account for farm density and proximity, globally considered among the most important drivers of disease spread [17,18]. For the geographic data manipulation and calculation, the R packages rgdal [19] and sp [20,21] were used. 

Infected farms belonging to the ‘same owner’, or close family members, were identified during the on-field epidemiological enquiries. The potential inter-outbreak contacts due to the possible unreported movements of the same owners were recorded into a symmetric dichotomic matrix.

The pairwise ‘genetic distances’ between the complete genomes of the viruses were calculated in MEGA 7.0.26 [22] and included in the analyses as the number of different nucleotides between the genotyped strains.

The list of all the network covariates included in the analysis is summarized in Table 1.

### 2.3. Statistical Analysis

The genomic and epidemiological information were combined to evaluate which drivers of infection could significantly explain the structure of the observed genetic network. Since a link between two nodes within the network represents the highest genetic similarity between viral strains, a potential lateral spread or common source of infection was assumed for the two infected farms from which the viruses were isolated. An exponential random graph model (ERGM) framework was developed to assess which potential risk factor can influence the probability of observing a link between two nodes, and, hence, might have significant effects of inter-farm transmission. The ERGM is a statistical model for network data that takes a generalized exponential family form and specifies the probability of an entire network as a function of covariates hypothesized to affect the network structure [23], and evaluates alternative hypotheses on the processes that could lead to the observed outcome [24]. For more detailed information on the exponential random graph modeling framework, its definitions, and mathematical notation, refer to [23,24,25,26].

In this study, the ERGM was used to perform a regression-like analysis, where the dependent variable is the presence (or absence) of a link between two nodes (i.e., two isolated viruses) while the explanatory variables are the characteristics of each outbreak and the dyadic relationships between pairs, as described above. The ERGM estimated coefficients can be interpreted in a similar fashion to those obtained from a logistic regression, and represent the change in the (log-odds) likelihood of a tie to occur, for a unit change in a predictor. The log-odds form of each predictor included in the analysis can be transformed into a corresponding probability (through the inverse-logit form), to ease the interpretation of how each variable affected the probability of observing ties between nodes. The model’s goodness of fit was assessed through the tie prediction statistic, which predicts the dyad states of the observed network by the dyad states of 1000 simulated networks. The tie prediction statistic includes the receiver-operating characteristics (ROC) and precision-recall (PR) curves [27].

All of the analyses were performed on the R software, using the following packages: ergm [25,28,29] for the model fit and analysis; btergm [30] for the calculus of the ties predicted probabilities; and ggplot2 [31], ggpubr [32], igraph [33], and ggraph [34] for the graphical representation of the genetic network and the results of the analysis.

## 3. Results

### 3.1. Evolution of the Italian Epidemiological Situation

A total of 317 HPAI H5N1 outbreaks in poultry farms and 23 in non-poultry birds were confirmed between 18 October 2021 and 1 April 2022 in nine Italian regions (Figure 1). The domestic outbreaks were located in seven regions, with Veneto and Lombardia being the ones where most cases were identified (Table 2). Fattening turkey farms (n = 150/317, 47.32%) were predominantly affected, as commonly observed in past events [4,5]; however, a strong involvement of the broiler sector was also registered, with about a quarter of the outbreaks notified in this production type (n = 77/317, 24.29%). The outbreaks reported in non-poultry birds were found in eight Italian regions spanning from Piemonte to the southern regions of Campania and Puglia (Figure 1, orange dots). Only 6 outbreaks out of a total of 23 involved birds belonging to the order Anseriformes (n = 6/23, 26.09%), which is considered among the target species for passive surveillance to detect AI in wild populations [7]. Most of the outbreaks in non-poultry birds affected seagulls (n = 6/23, 26.09%), birds of prey (n = 4/23, 17.39%), owls (n = 4/23, 17.39%), herons (n = 2/23, 8.70%), and magpies (n = 2/23, 8.70%). Some of the outbreaks in wild birds involved wildlife rescue centers, with cases in more than a single species.

The epidemic showed a rapid evolution, with a steady increase in the number of weekly cases until the second week of December, peaking at 51 cases per week. During this period, most outbreaks were notified in Veneto and Lombardia, while rare incursions were detected in other regions. From the third week of December, there was a sharp decline in the number of newly reported cases (Figure 2), and the latest were sporadic outbreaks confirmed in areas outside the area of major viral circulation, confirmed between mid-March and early April, in two rural farms in Toscana and Emilia-Romagna.

### 3.2. Genetic Analyses

We characterized the complete genome of 342 HPAI H5N1 viruses collected from wild (n = 21) and domestic (n = 321) birds for the outbreaks reported in Italy between October 2021 and April 2022; in some cases, more than a single isolate per outbreak was sequenced. All the viruses belonged to the clade 2.3.4.4b [35], and were clustered with viruses that had been circulating in Russia and Europe since the end of 2020. The phylogenetic analyses of the eight gene segments showed that the Italian HPAI H5N1 viruses belonged to six distinct genotypes originating from different reassortment events (Figure 3).

The genetic clustering suggested the occurrence of multiple viral introductions in the country, of which at least seven were detected in domestic birds (marked with different colours in the HA phylogenetic tree, Figure 3b). Most of the viruses collected from poultry outbreaks fell within three major clusters (in pink, blue, and green in Figure 3b), suggesting a sustained virus spread among poultry farms after the incursion of the virus into the domestic sector. Specifically, the genetic group, which includes most of the Italian viruses (pink cluster in Figure 3b), was the most widespread in northern Italy and was detected in several provinces of the Veneto and Lombardia regions (Figure 4), as well as in Poland and the Czech Republic (Appendix A). To elucidate the possible transmission dynamics among the affected farms, the viruses belonging to the pink cluster were subjected to a more in-depth study by using the phylogenetic network analysis based on the complete genome sequences of 229 Italian strains (Appendix A). The viruses showed a clear clustering by province, which might be indicative of the emergence of several secondary outbreaks caused by viral spread among the poultry farms. However, the occurrence of multiple introductions of genetically related viruses from wild birds into the domestic population cannot be completely excluded. The available genetic data make it impossible to distinguish the new introductions from wild birds from secondary spreads. Therefore, the identification of viruses with similar genetic constellations in other countries, as well as the presence of long branches connecting poultry outbreaks with no evident epidemiological connection, supported the hypothesis that this cluster could have been generated as a consequence of more than one virus incursion from wild birds into the domestic sector.

To further investigate the possible lateral transmission dynamics among the affected farms, a subset of the viruses belonging to the phylogenetic network was considered to construct a new genetic network (cluster A, Figure 5). This subset of sequences was selected because they belong to the largest group in terms of domestic outbreaks isolates and available epidemiological data. Moreover, the various genetic groups have markedly different phylogenetic characteristics, suggesting that they might have different epidemiology and transmission dynamics.

The cluster A network contains 249 nodes (214 of which were individual viral strains isolated from single Italian domestic outbreaks belonging to the larger genetic group, while 35 are the estimated median vectors), and 345 edges. The 214 outbreaks were confirmed between October and the end of December 2021, and they were mainly distributed in the Veneto Provinces (3 in Rovigo, 32 in Padova, 17 in Vicenza, and 158 in Verona), while only 4 were located in the Lombardia Region (2 in Cremona and 2 in Mantova provinces), as shown in Figure 4.

### 3.3. Epidemiological Analysis of the Genetic Network

The epidemiological data collected for the domestic outbreaks belonging to the genetic cluster A were summarized through an initial exploratory analysis. The contact tracing operations identified a total of 58 at-risk contacts through the movement of personnel and feed trucks, which involved 46 farms (up to a maximum of four at-risk contacts per farms dyad). Forty-five farms were found to belong to the same owners or relatives in the cluster included in the analysis, with up to six premises per single farmer. A total of 26 poultry companies, in addition to the rural sector, were affected by the end of the epidemic, and the one most affected had 108 outbreaks. All of the species and productive types were represented in cluster A: breeders (n = 3/214, 1.40% of the total outbreaks in the cluster), broilers (n = 55/214, 25.70%), laying hens (n = 27/214, 12.62%), fattening turkeys (n = 114/214, 53.27%), multispecies (n = 6/214, 2.80%), and other species (n = 9/214, 4.21%). The geographic distances between the infected farms ranged from a minimum of 130 m to a maximum of approximately 126 km, with a median of 22 km (interquartile range—IQR: 14–35). Lastly, the viral genetic differences ranged from a minimum of 0 (identical viral strains) to a maximum of 163 different bases, with a median of 21 (IQR: 13–27) bases.

The ERGM output can be similarly interpreted as the results that would be obtained from a logistic model. The coefficients are the change in the (log-odds) likelihood of a tie for a unit change in a predictor. Each log-odds of a tie can be transformed, through the logistic function, into a corresponding probability, considering the additive effect of each coefficient included in the model. The coefficient estimate for the ‘edge’ term, which represents the network density, is negative (Est._log-odds_ = −3.856, C.I._log-odds_ = [−4.009; −3.702]), indicating a much lower probability of observing a tie than expected in a network of the same size (tie baseline probability of 0.021). This can be interpreted as the fact that the observed network owns an intrinsic structure, in which the presence of ties between nodes (i.e., highest similarity between different isolated viral strains) can be explained by the influence of other variables. Four out of the seven predictors in the ERGM were significantly associated with the presence of network ties (Table 3). The ‘poultry company’ and ‘risk time windows’ variables were positively associated with a +13.4% higher probability of observing a tie when two farms belonged to the same company (Est._log-odds_ = 0.548, C.I._log-odds_ = [0.179; 0.918]), and +8.4% for each additional day of exposure (Est._log-odds_ = 0.339, C.I._log-odds_ = [0.309; 0.368]), respectively. Conversely, the ‘genetic differences’ and ‘geographic distance’ variables were negatively associated with the probability of observing two nodes connected within the genetic network. Specifically, a 13.7% lower probability was calculated for each additional different genomic base in the sequenced strains (Est._log-odds_ = −0.563, C.I._log-odds_ = [−0.640; −0.486]), and −1.4% for each additional kilometer of distance between farms (Est._log-odds_ = −0.058, C.I._log-odds_ = [−0.078; −0.038]). No significant effects were associated with the ‘species’, ‘at-risk contacts’, and ‘same owners’ variables.

The predicted probability distributions of the network ties are graphically presented in Figure 6. Specifically, the predicted tie probabilities are illustrated according to the three edge-level covariates that were shown to be significant by the model analysis (i.e., the ‘geographic distances’, the ‘risk time windows’, and the ‘genetic differences’), and differentiated according to the actual presence/absence of the ties within the genetic network. The probabilities estimated for the linked nodes within the genetic network are higher overall compared to the ones obtained for the unlinked nodes, which is indicative of good model performance. The predicted probabilities are the highest when the between-farm distances are below 1 km (median 0.35, IQR: 0.13–0.71), and when the exposure is extended to the maximum period of 15–16 days (median 0.56, IQR: 0.33–0.69, and 0.56, IQR: 0.37–0.74, respectively). As expected, the estimated ties’ probabilities according to the genetic distances tend to decrease as the number of different genetic bases increases, reaching the lowest median value between the 6 and 8 bases (median 0.00, IQR: 0.00–0.09, and 0.00, IQR: 0.00–0.02, respectively).

The model goodness of fit, intended as the reliability of the tie prediction, was evaluated through the calculation of the ROC and PR curves and their related performance metrics (Figure 7). The area under the ROC estimated for the model was 0.957, indicating a very good performance compared to the estimated baseline of a random classifier. However, as the ROC curve can be affected by a data imbalance (i.e., number of tie presences vs. absences) and it is threshold-invariant (i.e., it does not take into account when the false positive and false negative errors are unequal), the evaluation of the PR curve is also important, to avoid overestimating the model’s performance. In this case, an area under the PR curve of 0.477 reflects a very good model performance, compared to the estimated baseline PR of 0.009 (which is indicative of a very imbalanced class distribution, i.e., the positive class prevalence is much lower compared to the negative one).

## 4. Discussion

The evolution of the epidemiological situation in 2021–2022 in Italy was comparable only to the HPAI H7N1 epidemic of 1999–2000 in terms of spread and number of cases confirmed in the DPPAs [36]. Similarly to the 2017–2018 HPAI H5N8 outbreak [4], there was no initial phase of introduction of a low-pathogenicity virus from wild to domestic, followed by adaptation to poultry and subsequent increase in its pathogenicity (as in the 1999–2000 epidemic). However, in 2017, the virus had initially circulated on the fringes of the DPPA with repeated and distinct introductions from the wild, and only in the second wave did it involve areas with a higher density of poultry farms, with limited lateral spread [4]. On the contrary, the situation in 2021 immediately appeared to be more severe, as the HPAI virus was likely introduced directly into the core of the area with the highest poultry density of Italy, contributing to the rapid spread of the disease into the poultry sector, firstly involving the Verona province and then the neighbouring provinces. Moreover, the H5N1 virus behaved differently from what was expected, as it strongly affected the broiler sector (normally considered a low-risk species for AI infection [37]) with very limited or completely absent symptomatology [38]. In contrast to the 2021–2022 HPAI A (H5N1) epidemic elsewhere in Europe, wild bird cases in Italy were rare in comparison to domestic cases. However, the role of wild birds in both epidemic persistence and geographical dispersal in Italy has been proven to be significantly greater than apparent in the past, when considering the wild bird hosts’ reconstruction in the phylogenetic tree [39]. This suggests that the passive surveillance activities foreseen under the current Italian national surveillance plan could be complemented by active sampling and testing of birds belonging to the orders Anseriformes and Charadriiformes, to enable transmission to be understood in greater detail and to provide an early warning for the possible introduction of the disease in poultry so that operators can apply reinforced preventive measures without delay.

The analysis presented in this study showed that there were significant effects of some characteristics, related to the domestic farms and the study area, on the probability that the viruses isolated from two different outbreaks were genetically connected. These associations could be explained by the multiple occurrences of lateral spreads, or the presence of common sources of infection, which plausibly occurred massively during the course of the 2021–2022 epidemic and which would explain such a large number of poultry farms infected in such a short period. The drivers associated with a higher risk of infection spread are the longer exposure to active outbreaks, the geographical proximity of poultry farms, and belonging to the same poultry company. The period in which farms remained exposed to the risk of infection (hypothesized to occur via at-risk contact through vehicles and personnel, or by proximity) due to the persistence of active outbreaks was found to be one of the most relevant determinants of the 2021–2022 epidemic’s evolution. The model showed that for each additional day that a farm was exposed to contact with an active outbreak, the risk of transmission increased by just over 8%. This finding is particularly relevant because it strongly indicates the need to extinguish the outbreaks as quickly as possible to reduce the risk of further transmission in an area characterized by a very high density of poultry premises. Hence, adequate preparedness to respond to epidemic emergencies is essential, including development of ad hoc protocols for more rapid and efficient extinction operations, such as culling, removal of carcasses, and disinfections, but measures that aim at reducing the population exposed to active outbreaks (e.g., through targeted pre-emptive culling) are also needed. The observed inverse relationship between the geographic distance and the risk of disease transmission between farms is consistent with the results found in the literature [40,41,42], and reflects the importance of reducing the farms density. Although the ban on restocking was applied within the restricted areas during the epidemic, the reduction in the number of operating farms could be included in a package of preventive measures following an increased risk of AIV introduction in the DPPAs. Such a reduction could be reached, for instance, through the temporary ban on restocking farms in the DPPA, or anticipating the slaughtering of animals approaching the end of the production cycle, in response to the outcome of early warning systems. The organization of poultry companies may play a major role in the evolution of the epidemiological situation, as differences in biosecurity standards and operating procedures may reflect different management capabilities, which in turn results in different risk levels of disease introduction. However, during the 2021–2022 epidemic, even large poultry companies were severely affected, indicating a massive environmental viral pressure that could have hampered the biosecurity measures already in place, and prompting further structural or managerial improvements or changes to prevent the recurrence of such a severe lateral transmission. A critical point identified during the last epidemic was related to the difficulties in collecting information on vehicles and personnel traceability, especially for smaller poultry companies. In fact, the variable ‘same poultry company’ can be considered as a proxy for this missing information, indicating the possibility of more frequent contact occurring between farms belonging to the same company than those documented. This variable could also include those at-risk contacts that were not properly reported in the visitor logbooks and that can be referred to each specific company (e.g., maintenance staff for water/gas/electrical systems, delivery of drugs for specific treatments, etc.). Finally, it is also possible that some procedures are typical to individual poultry companies and/or productive types (e.g., companies specializing in organic production, free-range egg production, free-range farming, etc.). These procedures could be linked to particular types of biosecurity management and, thus, to a different risk of AIV transmission. Similar results were also discussed in the study of Yoo and colleagues [40], in which the variables ‘shared poultry integrators’ and ‘closer geographical distance’ were identified as significant contributing factors for potential transmission routes between premises. The variables related to the number of at-risk contacts, the same reared species, and belonging to the same owner were not significantly associated with the genetic similarity structure observed in the network, nor, thus, to the probability of lateral spreads. Several other studies evaluated the impact of potentially at-risk movements on the probability of AI spread among farms, and most of them report a significant association between the two. For example, some studies conducted on the 2016–2017 H5N8 epidemic in France found that live-duck movements and indirect contacts via the transit of transport vehicles might have played a crucial role in the AI diffusion. According to Guinat et al. [42], live-duck movements might have played a crucial role in the geographical spread of AIV; although, as reported by Chakraborty et al. [43], viral spread was mostly related to short-range movements, i.e., between neighboring or within the same administrative regions. The analysis presented in the work of Bauzile et al. [44] also showed that transmission events among farms were likely to have occurred through live animal movements and the transit of transport vehicles. In Yoo et al. [40], an extensive list of 18 different types of movements was found to be associated with a higher risk of transmission. The reason for the discrepancy between the present study and those reported in the literature could be related to some differences in the traceability systems and logistics organization between countries. For example, the Italian traceability system of feed trucks and personnel movements relies, at the present time, on the collection of non-standardized data directly from the poultry integrators that assemble transport notes, as well as information from personal diaries to reconstruct the taken routes. The reliability and accuracy of this data collection system are not comparable, for example, to the one used in the Republic of Korea, which is based on satellite tracking using global positioning system technology [40], or the official registration system that is mandatory for livestock movements. In addition, in this study, only contacts related to the movements of feed trucks and poultry company personnel could be retrieved for analysis, and these data represent only a part of all the movements that normally take place from and into farms. The number of identified at-risk contacts potentially associable with lateral spread events (n = 58) was, indeed, markedly low considering the total number of retrieved movements among farms (n = 1470), representing only 3.94% of the overall movements, potentially explaining the non-significance of this variable in our analyses. Nevertheless, it is important to take into account the severe risk associated with the presence of even a few contacts, which should not occur in the course of an HPAI epidemic. The presence of these potentially problematic movements might indicate the presence of breaches in the organization of transport logistics, prompting a more suitable organization of vehicle movements to prevent multiple contacts of the same lorries with different farms. At the same time, it is recommended that the biosecurity measures be strengthened at the farm level, and that the actual effectiveness of the disinfection procedures to and from the farms be evaluated. 

Rearing the same species or having the same production type, as well as belonging to the same owner or a member of the same family, were also not significantly associated with the lateral spread events. Interestingly, in Yoo et al. [40] the variable accounting for the similarities in the reared poultry species resulted to be related to a higher risk of transmission. This difference might be due to the effect of the ‘geographical distance’ variable. In the Italian DPPAs, the probability that farms in close proximity breed the same species or have the same owner is very small, indicating that the geographic distances are likely more strongly correlated with the risk of viral spread between outbreaks, thus hedging the effect of the other two variables. Moreover, from the information collected during the 2021–2022 epidemic, the same feed truck was reported as supplying up to five different poultry production types on the same day.

Lastly, although the variable related to the number of different bases between two viral strains does not have a typical epidemiological value, its inclusion in the model analysis was useful to estimate, on average, the magnitude of the viral genomic diversity necessary to consider two strains arising from two separate origins, even if belonging to the same cluster. The probability of observing two viruses connected within the genetic network (and thus, a higher chance of direct transmission between farms, or a common source of infection), decreases by 13.7% for each additional different base. Taking into account the characteristics of the H5N1 virus circulating in Italy in 2021–2022, the results indicate that at least 7–8 nucleotide substitutions might be sufficient to discriminate outbreaks related to each other from those that were not connected.

The analysis of risk factors for potential lateral spread presented here emphasized the importance of prompt interventions following confirmation of the outbreak, leading to its rapid extinction and possibly reducing the density of the farms at the local level. However, the high incidence of the disease, especially during the first weeks of the epidemic, led to a rapid saturation of the farms’ culling and disposal capacity. This resulted in numerous active outbreaks in close geographical proximity, a combination that led to a further massive spread of the virus. An ideal solution to address this critical issue would be a temporary reduction in the farms’ density, e.g., by restricting or banning the restocking of the most susceptible species in defined areas of the highest risk zones, in response to an adequately structured early warning system. Other hypotheses, regarding the effect of transports to the slaughterhouse of potentially infected animals in spreading the virus to the farms located in close proximity to the routes, cannot be ruled out, and would require detailed data collection (e.g., information on the routes taken by the trucks) and/or analytical methods based on complex mathematical simulations. The role played by the broiler species in the AI dynamics is a further crucial point to investigate, as they showed very few or no symptoms in the course of infection [38], indicating that the time window to consider for the viral introduction should be carefully recalculated for evaluation of the at-risk period of interest for the contact tracing activities.

In addition to the assessment of the effect of implementing further control measures to counter the AI spread, especially in DPPAs, additional analyses are also necessary to gain knowledge on the rapidly changing epidemiology of AI over the last decade. These include, for example, adding temporal information to identify any significant fluctuation in the effect of the risk factors during the epidemic. Another interesting approach would include a Bayesian analysis to quantify and consider the effect of uncertainty associated with the potential epidemiological drivers, as well as reconstructing the missing information related to the median vector nodes estimated during the generation of the genetic network. Mathematical models would be useful to investigate the AI spread, to calculate the actual reproductive number (Rt) during the course of the epidemic, and to evaluate which control measures may have contributed the most to reducing Rt below the critical threshold necessary for spreading events. Finally, the results of the ERGM could be integrated with other mathematical models, e.g., based on an epidemic-tree approach, to reconstruct the most likely route of transmission that occurred within the population and to simulate how the modulation of specific measures (e.g., preventive culling within a defined radius, farm blocking, stamping out of at-risk contact farms, vaccination, etc.) may affect the transmission dynamics. In particular, the potential effect of vaccination would be of interest, as it could contribute to reducing the risk of between-farm spread of the virus, as well as minimizing the preventive culling of large numbers of healthy birds and, ultimately, human exposure to the AI viruses, while proving an HPAI-free status for trading purposes [45]. Moreover, while reaffirming that biosecurity and surveillance remain important cornerstones of the fight against HPAI, the EU Council and Member States are now considering that vaccination against HPAI, focused on at-risk areas, species, and farming practices, could become a relevant complementary tool for the prevention and control of the most serious epizootic in recent history in Europe.

## Figures and Tables

**Figure 1 pathogens-12-00100-f001:**
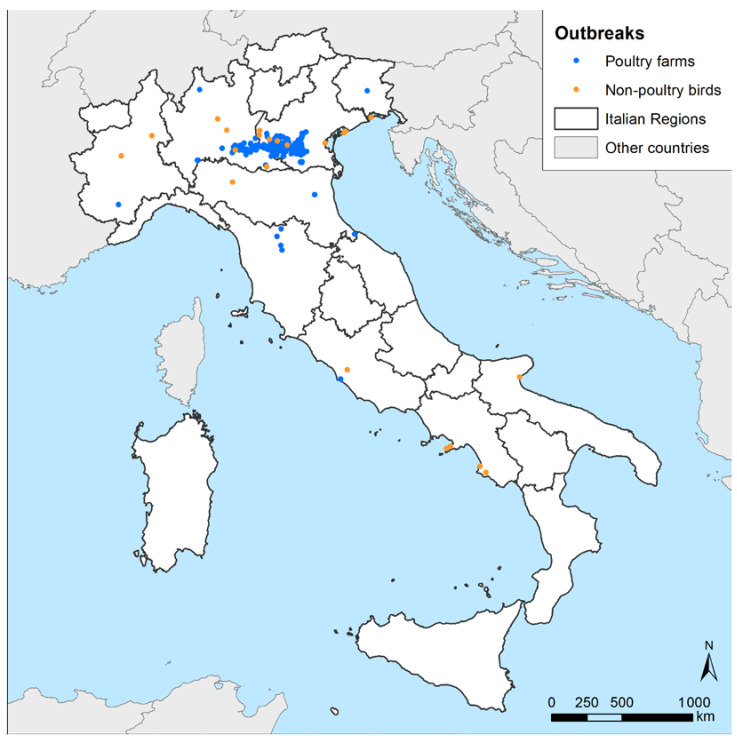
Notified Italian HPAI H5N1 outbreak distribution in poultry farms and non-poultry birds (22 in wild birds and 1 in captive birds) during the 2021–2022 epidemic.

**Figure 2 pathogens-12-00100-f002:**
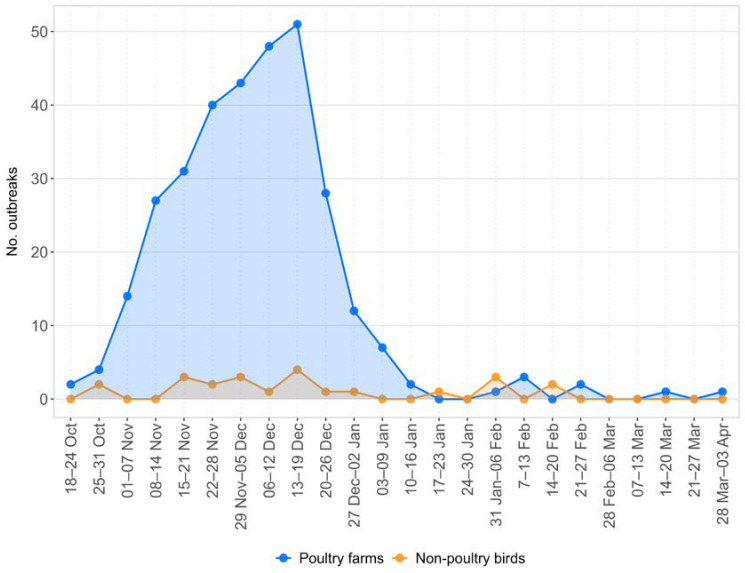
The epidemic curve showing the number of weekly cases in poultry farms and non-poultry birds (22 in wild birds and one in captive birds).

**Figure 3 pathogens-12-00100-f003:**
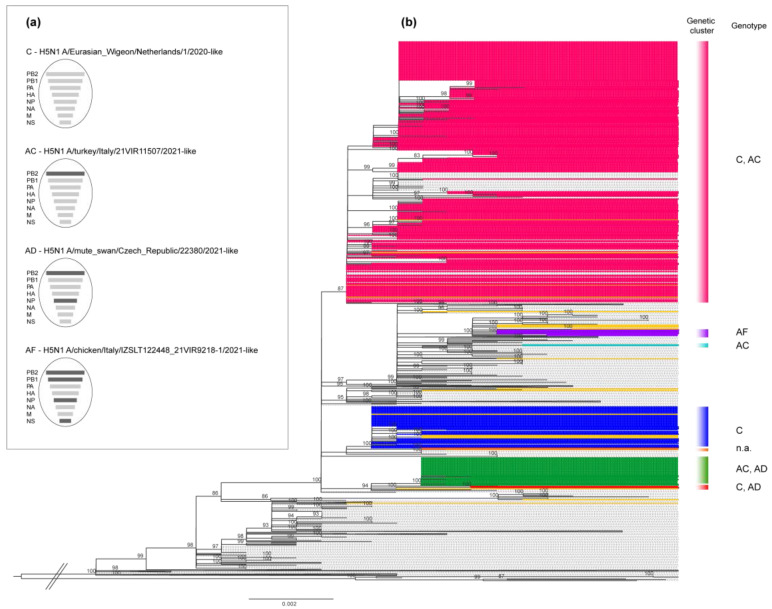
Phylogenetic tree of the hemagglutinin (HA) gene segment and genotypes identified among the Italian H5N1 viruses. (**a**) Schematic representation of the gene composition of each genotype. Light grey bars: conserved segments; dark grey bars: gene segments acquired by reassortment events. (**b**) Phylogenetic tree of the HA gene segment colored according to seven different genetic clusters. C, AC, AD, AF: different genotypes characterizing the Italian viruses; n.a.: viruses not assigned to genotypes due to the absence of one or more gene segments; pink cluster: genetic group that includes most of the Italian viruses collected from October 2021 to April 2022; dark yellow sequences: viruses collected from wild birds.

**Figure 4 pathogens-12-00100-f004:**
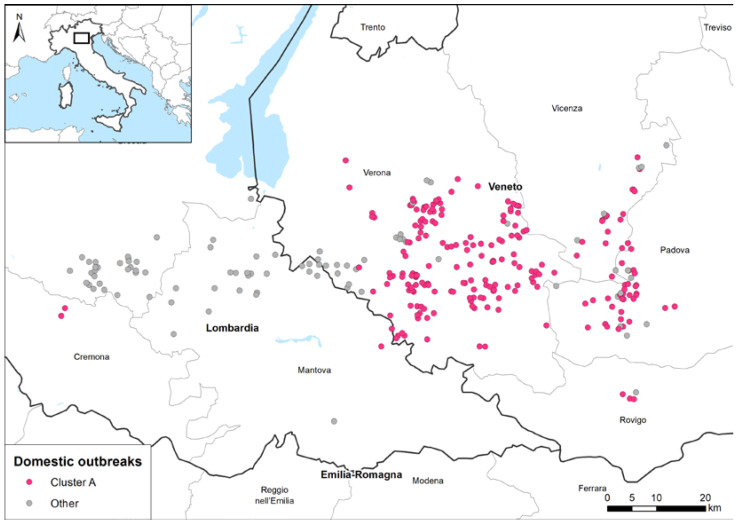
Geographic distribution of the domestic outbreaks in a densely populated poultry area. Pink dots: cluster A outbreaks; grey dots: non-clustered outbreaks or outbreaks belonging to other minor genetic groups.

**Figure 5 pathogens-12-00100-f005:**
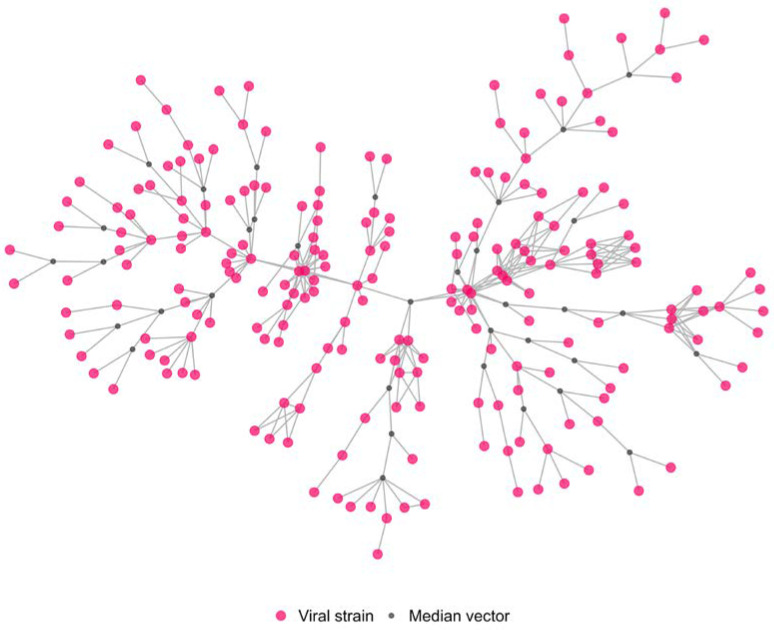
Graphical representation of the estimated genetic network (cluster A). Pink nodes: H5N1 HPAI virus isolated from individual domestic outbreaks belonging to the larger genetic group; grey nodes: estimated median vectors; links: highest genetic similarity between nodes.

**Figure 6 pathogens-12-00100-f006:**
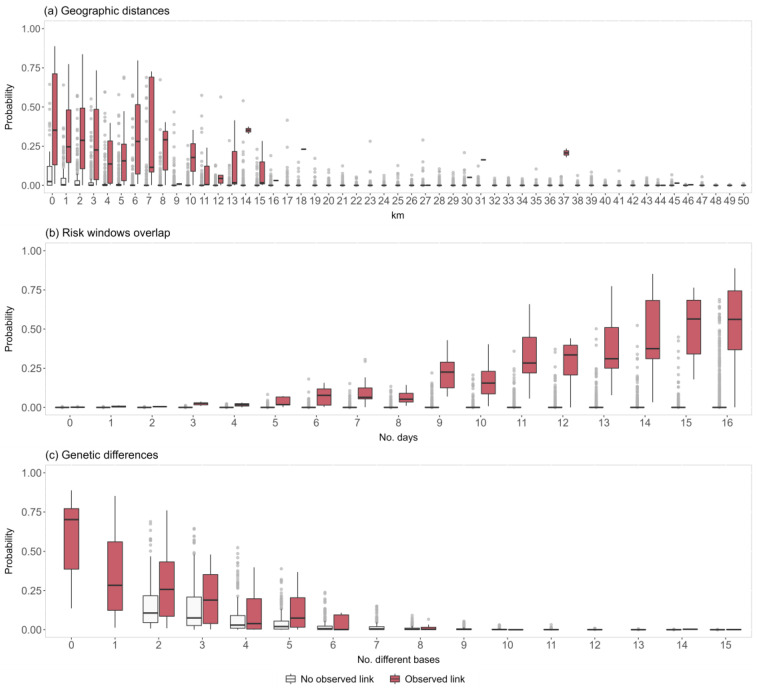
Graphical representation of the predicted probability distributions according to the (**a**) geographic distances, (**b**) risk time windows overlap, and (**c**) viral genetic differences. The *x*-axis scale for (**a**,**c**) was limited to 50 km and 15 bases, respectively, to improve the graph’s readability.

**Figure 7 pathogens-12-00100-f007:**
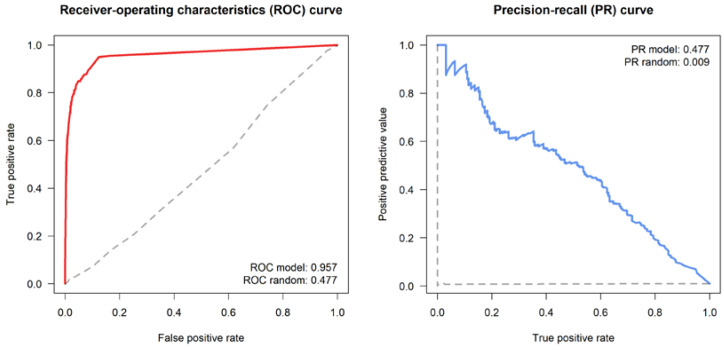
Tie prediction statistics, including the ROC (**left panel**) and PR (**right panel**) curves. Solid lines: actual ROC/PR curves; dashed lines: baseline ROC/PR curves drawn for a random graph.

**Table 1 pathogens-12-00100-t001:** List of network covariates included in the analysis.

Type	Name	Description
Density	Edges	The overall network density considering the total number of observed ties
Nodal-level covariates	Species	Network homophily by species
	Poultry company	Network homophily by poultry company
Edge-level covariates	Risk windows overlap	Number of overlapping days considering the out-bound and the in-bound risk time windows
	At-risk contacts	Number of potentially productive contacts occurred between farms
	Geographic distances	Farms proximity calculated as Euclidean distance (m)
	Same owner	Presence/absence of an owner of multiple infected farms
	Genetic differences	Number of different bases between the isolated strains

**Table 2 pathogens-12-00100-t002:** Avian influenza outbreaks in poultry farms by region.

Region	Fattening Turkey	Broiler ^1^	Laying Hen ^2^	Breeder^3^	Multi-Species ^4^	Other Species ^5^	Total
Emilia-Romagna					2		2
Friuli-Venezia Giulia		1					1
Lazio					1		1
Lombardia	20	9	20		5	6	60
Piemonte		1					1
Toscana					4		4
Veneto	130	65	32	5	7	9	248
Total	150	77	51	5	17	15	317

^1^ Chicken broilers and cockerels; ^2^ pullets and ready-to-lay hens; ^3^ guinea fowl breeders, chicken breeders and turkey breeders; ^4^ rural/backyard farms and agritourism; ^5^ quails, pheasants, and guinea fowl.

**Table 3 pathogens-12-00100-t003:** Coefficient estimates, standard errors (S.E.), 95% coefficients intervals, *p*-values, and corresponding probabilities for the explanatory variables included in the ERGM analysis.

Variable	Coefficient Estimate (Log-Odds)	S.E.	C.I. [2.5%; 97.5%]	*p*-Value ^1^	Corresponding Probability (Inverse Logit)	Probability vs. Randomness
Edges	−3.856	0.078	[−4.009; −3.702]	***	0.021	−47.9%
Species	0.218	0.179	[−0.132; 0.568]			
Poultry company	0.548	0.188	[0.179; 0.918]	**	0.634	+13.4%
At-risk contacts	−0.062	0.591	[−1.222; 1.096]			
Same owners	0.718	0.603	[−0.465; 1.900]			
Risk time windows	0.339	0.015	[0.309; 0.368]	***	0.584	+8.4%
Genetic differences	−0.563	0.039	[−0.640; −0.486]	***	0.363	−13.7%
Geographic distances	−0.058	0.010	[−0.078; −0.038]	***	0.486	−1.4%

^1^*p*-value: ***, *p* < 0.0001; **, *p* < 0.001.

## Data Availability

Data sharing not applicable due to privacy restrictions.

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
