# Peer review of "Integration of Epidemiological and Genomic Data to Investigate H5N1 HPAI Outbreaks in Northern Italy in 2021–2022"

_pathogens, 2023, doi:10.3390/pathogens12010100_

Round 1
Reviewer 1 Report
Possible causes of the dynamic spread of HPAIV H5N1 in Italian poultry premises in 2021-22 are investigated. Extensive sequence information and detailed field epidemiological data are used to inform an exponential random graph model. The ERGM is established and validated here and meaningful conclusions are drawn from its output. These indicate that dynamic transmission between premises was a major factor of lateral spread. Several risk factors were identified that fostered spread. Among them “shared ownership of holdings” and the “size of overlap of a risk window” showed strong linkages with spread. The authors conclude that useful counteractive measures are based on rapid outbreak identification and immediate removal of the infectious holding. However, due to the large numbers of outbreaks in a short period, culling capacities were rapidly suffocated. Instead, the authors state, cuts into the population structure of susceptible poultry aiming at a reduction of population density should be encouraged.
This is a very interesting study exemplifying how to connect genetic, phylogenetic and epidemiologic data in a meaningful and profitable sense.
There are just two remarks that should be considered:
1. Did the authors obtain any data on the transmission between holdings with different poultry species; e.g. was turkey-2-turkey more easily achieved compared to e.g. turkey-2-duck?
2. In their list of measures they suggest towards the end of the discussion this reviewer misses vaccination as a further tool to reduce Ro of between-farm spread. A short paragraph, maybe as an outlook, should still be added.
Minor suggestions:
12-20 The different factors should be indicated by apostrophs, `same owner´, `genetic difference´etc.
21 Consider to replace “suggesting” by “casting light on”
23 “reorganization” is a wide field… Could this be rephrased a bit more specific as to which kind of organization the authors favor?
43 “bounded” or “narrowed” instead of “limited”?
215-8 The list of affected wild bird species seems to indicate that mainly scavengers have been found. These birds may have become infected by preying/scavenging on carcasses of the truly source/species of incursion to Italy?
Reviewer 2 Report
In the manuscript entitled, "Integration of Epidemiological and Genomic Data to Investigate H5N1 HPAI Outbreaks in Northern Italy in 2021-2022" the authors studied the association of epidemiological data and genomic analyses data of the 2021-2022 outbreak in Italy to monitor the epidemic evolution and transmission dynamics. Generally, the manuscript addresses an interesting topic and it is well-organized, and well-written.
Line 54: “a total of 317 outbreaks were notified....... “Please provide a reference (s). Same at line 63.
Line 626: reference 21 looks incomplete.
